# The Phosin PptA Plays a Negative Role in the Regulation of Antibiotic Production in *Streptomyces lividans*

**DOI:** 10.3390/antibiotics10030325

**Published:** 2021-03-20

**Authors:** Noriyasu Shikura, Emmanuelle Darbon, Catherine Esnault, Ariane Deniset-Besseau, Delin Xu, Clara Lejeune, Eric Jacquet, Naima Nhiri, Laila Sago, David Cornu, Sebastiaan Werten, Cécile Martel, Marie-Joelle Virolle

**Affiliations:** 1Institute for Integrative Biology of the Cell (I2BC), Université Paris-Saclay, CEA, CNRS, 91198 Gif-sur-Yvette, France; nshikura@k4.dion.ne.jp (N.S.); emmanuelle.darbon@i2bc.paris-saclay.fr (E.D.); catherineesnault@neuf.fr (C.E.); xudelin@live.com (D.X.); lejeune.clara@orange.fr (C.L.); Laila.SAGO@i2bc.paris-saclay.fr (L.S.); David.CORNU@i2bc.paris-saclay.fr (D.C.); cecile.martel@i2bc.paris-saclay.fr (C.M.); 2Laboratoire de Chimie Physique (LCP), CNRS UMR 8000, Université Paris-Saclay, 91405 Orsay, France; ariane.deniset@gmail.com; 3Department of Ecology, Institute of Hydrobiology, School of Life Science and Technology, Key Laboratory of Eutrophication and Red Tide Prevention of Guangdong Higher Education Institutes, Engineering Research Center of Tropical and Subtropical Aquatic Ecological Engineering, Ministry of Education, Jinan University, Guangzhou 510632, China; 4Institut de Chimie des Substances Naturelles, CNRS, Université Paris Saclay, 91190 Gif-sur-Yvette, France; Eric.JACQUET@cnrs.fr (E.J.); Naima.NHIRI@cnrs.fr (N.N.); 5Institute of Biological Chemistry, Biocenter, Medical University of Innsbruck, Innrain 80, 6020 Innsbruck, Austria; Sebastiaan.Werten@i-med.ac.at

**Keywords:** phosphate, polyphosphates, CHAD domain, antibiotic, pho regulon

## Abstract

In *Streptomyces*, antibiotic biosynthesis is triggered in phosphate limitation that is usually correlated with energetic stress. Polyphosphates constitute an important reservoir of phosphate and energy and a better understanding of their role in the regulation of antibiotic biosynthesis is of crucial importance. We previously characterized a gene, *SLI_4384/ppk*, encoding a polyphosphate kinase, whose disruption greatly enhanced the weak antibiotic production of *Streptomyces lividans*. In the condition of energetic stress, Ppk utilizes polyP as phosphate and energy donor, to generate ATP from ADP. In this paper, we established that *ppk* is co-transcribed with its two downstream genes, *SLI_4383*, encoding a phosin called PptA possessing a CHAD domain constituting a polyphosphate binding module and *SLI_4382* encoding a nudix hydrolase. The expression of the *ppk/pptA/SLI_4382* operon was shown to be under the positive control of the two-component system PhoR/PhoP and thus mainly expressed in condition of phosphate limitation. However, *pptA and SLI_4382* can also be transcribed alone from their own promoter. The deletion of *pptA* resulted into earlier and stronger actinorhodin production and lower lipid content than the disruption of *ppk*, whereas the deletion of *SLI_4382* had no obvious phenotypical consequences. The disruption of *ppk* was shown to have a polar effect on the expression of *pptA*, suggesting that the phenotype of the *ppk* mutant might be linked, at least in part, to the weak expression of *pptA* in this strain. Interestingly, the expression of *phoR/phoP* and that of the genes of the *pho* regulon involved in phosphate supply or saving were strongly up-regulated in *pptA* and *ppk* mutants, revealing that both mutants suffer from phosphate stress. Considering the presence of a polyphosphate binding module in PptA, but absence of similarities between PptA and known exo-polyphosphatases, we proposed that PptA constitutes an accessory factor for exopolyphosphatases or general phosphatases involved in the degradation of polyphosphates into phosphate.

## 1. Introduction

*Streptomyces* are Gram-positive, filamentous soil bacteria, of great medical and economic importance since they are able to produce a great variety of bio-active molecules useful to human health or agriculture [1,2]. The biosynthesis of these molecules usually takes place when growth slows down or stops and is thought to be triggered by some nutritional limitation [3]. One of the most efficient triggers of antibiotic production is a limitation in phosphate that is correlated with a low ATP content (weak energetic state) [4,5]. Conversely, antibiotic production is strongly repressed in condition of high phosphate availability. Inorganic phosphate being often a limiting nutriment in the environment, living systems have developed ways to store it efficiently as polyphosphates, in stressful conditions [6,7]. Polyphosphates (polyP) are linear polymers of phosphate linked by highly energetic phosphoanhydride bonds. Their synthesis relies, at least in part, on the action of polyphosphate kinases, enzymes able to polymerize the γ phosphate of ATP into polyphosphate [8]. However, since some polyP were still present in the *ppk* mutant of *E. coli* [9], alternative mechanisms of polyP synthesis are likely to exist. The accumulation of large quantities of polyP was noticed in *phoU* mutants of *E. coli* [10] and was correlated with an activation of Pi up-take by the PstSCAB transport system. This suggested that active Pi up-take contributes to polyP synthesis. Polyphosphates are involved in the resistance to various stresses [11,12,13] and constitute an important intracellular reserve of phosphate and energy that can be mobilized in condition of phosphate and energetic stress [14]. In such conditions, polyphosphates can either be used as a phosphate and energy donor to generate ATP from ADP by enzymes possessing adenosine diphosphate kinase-like activities, such as Ppk (SCO4145/SLI_4383) in *Streptomyces* [15,16] or as a an inorganic phosphate supplier via its degradation by endo and/or exo-polyphosphate phosphatases. In *Streptomyces* species, antibiotic biosynthesis being triggered in condition of phosphate limitation, getting a better understanding of the role played by this important polymer in the regulation of antibiotic biosynthesis is of crucial importance.

We previously characterized the polyphosphate kinase (Ppk) encoding gene of *S. lividans* (*SLI_4384*) and demonstrated that, *in vitro*, Ppk can act either as a polyphosphate kinase, synthetizing polyphosphate from ATP, or as an adenosine diphosphate kinase, regenerating ATP from ADP and polyP, depending on the ATP/ADP ratio in the reaction mix [15]. In *S. lividans*, *ppk* is mainly expressed in Pi limitation and belongs to the Pho regulon [16,17]. *In vivo*, in Pi limitation, the polyP and ADP content of the *ppk* mutant was shown to be higher than that of the *wt* strain, suggesting the absence of a polyP-dependent regeneration process of ADP into ATP [5]. However, unexpectedly the ATP content of the strain was similar to that of the *wt* strain, suggesting that homeostatic processes, similar to those occurring in *S. coelicolor*, are being triggered in this strain in order to re-establish its energetic balance [5]. Consistently, in Pi limitation, the deletion of *ppk* led to a huge increase of the weak antibiotic production of *S. lividans* that was correlated with a lower total lipid content, as in *S. coelicolor* [5,15].

Interestingly, the proteins encoded by the two genes located immediately downstream of *ppk* have similarities with proteins possibly involved in polyP metabolism. *SLI_4383/pptA* encodes a protein bearing a conserved histidine α-helical domain (CHAD domain, PFAM PF05235). Proteins with CHAD domain are usually found associated with polyP granules in various organisms and are called phosins (Tumlirsch and Jendrossek 2017; Lorenzo-Orts, et al., 2019). The CHAD domain is thought to constitute a positively charged tunnel accommodating negatively charged polyP [18,19,20]. *SLI_4382* encodes a protein of the nudix hydrolase family bearing similarities with enzymes involved in the cleavage of diadenosine hexaphosphate (Ap6A) into two molecules of ATP. The transcriptional situation of these three genes was determined in condition of phosphate limitation and proficiency in the wild type strain of *S. lividans*, in a *phoP* deletion mutant of this strain as well as in *S. coelicolor*. The disruption of *SLI_4383/pptA* and *SLI_4382* was achieved in order to determine whether the corresponding proteins play a role in the regulation of antibiotic biosynthesis. This study unexpectedly revealed that *pptA* was weakly expressed in the *ppk* mutant and that its disruption had a higher impact on antibiotic production and total lipid content than the disruption of *ppk*. This suggested that the phenotype of the *ppk* mutant might be due, at least in part, to the weak expression of *pptA* in this strain.

## 2. Results

### 2.1. SLI_4382 and SLI_4383 Amino Acid Sequence Features

At the Sanger center, SLI_4382 is described as a 142 AA long protein belonging to the nudix hydrolase family and bearing similarities with enzymes involved in the cleavage of diadenosine hexaphosphate (Ap6A) into two molecules of ATP.

On its side, SLI_4383/PptA is described as a 388 AA long protein bearing a TTA leucine codon, possible target for *bldA* regulation [21] and possessing a conserved histidine alpha-helical domain (“CHAD” domain), considered as a polyphosphate binding module [19,20]. Interestingly, a comparison of the predicted protein product of *SLI_4383*/*pptA* (GenBank entry EFD67904.1) to putative orthologues from other *Streptomyces* species revealed the presence of N-terminal extensions of various length whose extremity is highly conserved in almost all orthologues but which is missing in the protein of *S. lividans* as it is annotated at the Sanger center (Figure 1).

In the recently determined crystal structure of *S. chartreusis* PptA [20], this N terminal region comprises part of the first α-helix of the protein fold. Since in the *S. lividans* genome (GenBank entry GG657756.1) a highly similar sequence is present immediately upstream of the designated *pptA* translational start codon, it seems likely that this codon was erroneously assigned to an internal GTG (Val) codon. Predictions using various programs listed in Materials and Methods further confirmed this hypothesis and indicated that the most probable in frame GTG start codon is located 68 codons upstream of that provided at the Sanger center (GenBank entry EFD67904.1). *S. lividans* PptA would thus be 456 AA long, with a coding region 204 nucleotides longer than anticipated and overlapping with the C-terminal region of *ppk*, the GTG start codon of *pptA* being located 17 nt upstream of the TGA stop codon of *ppk*.

### 2.2. Ppk, pptA and SLI_4382 Are Co-Transcribed But pptA and SLI_4382 Can Also Be Transcribed Alone from Their Own Promoter

In order to establish the transcriptional situation of the genes *ppk*, *pptA*, and *SLI_4382* that are located just upstream of the genes encoding the high affinity phosphate transporter (PstSCAB), RNA was prepared from *S. lividans* grown for 40 h on solid R2YE medium with no phosphate added (1 mM final, Pi limitation) and specific primers, internal to each gene or between two adjacent genes, were designed to assess their transcriptional organization by RT-qPCR (Figure 2a).

Amplification of fragments, at a similar level, was obtained with primers A and B (internal *ppk*) and primers C and D (*ppk/pptA* junction) as well as with primers E and F (internal to *pptA*) and primers G and H and primers I and J, corresponding to the *pptA/SLI_4382* junction and internal to *SLI_4382*, respectively (Figure 2b). These results demonstrated that these three genes constitute an operon.

The co-transcription of *ppk* and *pptA* led to the prediction that the disruption of the *ppk* gene by a resistance cassette might have a polar effect on *pptA* expression. In order to test this hypothesis, RNA was prepared from *S. lividans* and its *ppk* mutant grown for 40 h on solid R2YE medium with no phosphate added (1 mM final, Pi limitation) and primers K and L were used to detect *pptA* expression. Results shown in Figure 2c indicated that *pptA* was still expressed in the *ppk* mutant but at a much lower level than in the *wt* strain. In consequence, considering the polar effect of the insertion of the hygromycin cassette in *ppk* on *pptA* expression, the antibiotic overproducing phenotype of the *ppk* mutant might be, at least in part, due to the weak expression of *pptA* in this strain.

This residual weak transcription of *pptA* in the *ppk* mutant suggested that *pptA* might also be transcribed alone from its own promoter. In order to test this hypothesis, RACE-PCR experiments were conducted to determine whether a transcriptional start point (TSP) could be detected upstream of the coding sequences of *SLI_4383/pptA* as well as of *SLI_4382* (Figure 3a,c). Indeed, a TSP located at a C, 13 nt upstream of the newly proposed GTG start codon of *pptA,* and thus 30 nt upstream of the Ppk TGA stop codon was identified. This allowed the positioning of putative −10 and −35 sequences of this gene at the end of the *ppk* coding sequence (Figure 3b). Furthermore, a TSP located at a C, 86 nt upstream of the GTG start codon of *SLI_4382,* and thus 37 nt upstream of the *pptA* TAG stop codon was also identified. This allowed the positioning of putative −10 and −35 sequences of this gene at the end of the *pptA* coding sequence (Figure 3d).

### 2.3. The ppk/pptA/SLI_4382 Operon Belongs to the Pho Regulon

In order to establish the impact of phosphate availability on the transcriptional regulation of the genes of this operon, RNA was prepared from *wt S. lividans*, its *phoP* mutant and *wt S. coelicolor* grown for 40 h on solid R2YE medium with no phosphate added (1 mM final, Pi limitation) or with 4 mM phosphate added (5 mM final, Pi proficiency).

Results of RT-qPCR shown in Figure 2d confirmed that the expression of these genes was induced in condition of phosphate limitation and repressed in condition of phosphate proficiency in the *wt* strain of *S. lividans*, whereas the expression of these genes was 20-fold lower in the *phoP* mutant and at a similar level in both Pi conditions (Figure 2d). This demonstrated that these genes belong to the Pho regulon. The expression of these genes was 4 to 5 fold lower in *S. coelicolor* than in *S. lividans*, in condition of Pi limitation (Figure 2d), consistently with the previously reported weak expression of genes of the *pho* regulon in *S. coelicolor* [22].

In condition of Pi proficiency, the expression of these genes was low in the three strains and slightly lower in the *phoP* mutant and in *S. coelicolor* than in the *wt* strain of *S. lividans* (Figure 2d). Interestingly, the expression of the first gene of the operon, *ppk*, was four-fold higher than that of its downstream genes. This intracistronic polarity is well documented in the literature and some studies suggest that it might be due to the degradation of inefficiently translated RNA [23]. The overlap of the 3′ end of *ppk* and 5′ end of *pptA* or of the 3′ end of *pptA* and 5′ end of *SLI_4382*, might reduce the translation efficiency of *pptA* and *SLI_4382* making their transcripts more susceptible to degradation.

### 2.4. The Disruption of SLI_4383/pptA Confers an Actinorhodin Over-Producing Phenotype to S. lividans TK24

In order to determine whether the disruption of *SLI_4383* had an impact on actinorhodin (ACT) production of *S. lividans*, *pptA* was replaced by antibiotic resistance cassette by PCR-targeting. The cassette was then excised as described in Materials and Methods. A similar disruption was carried out for *SLI_4382* but the disruption cassette was not excised.

On solid R2YE medium in Pi limitation, the *pptA* mutant was characterized by an enhanced ACT production, whereas the *SLI_4382*::*att3-aac* mutant had a phenotype identical to that of the *wt* strain (Figure 4a). The assay of ACT production revealed that the *pptA* mutant strain produced ACT earlier (60 h versus 72 h) and in 1.5- to 2-fold higher amount than the *ppk* mutant at 72 h. ACT production of the *pptA* and *ppk* mutant strains reached their maximum at 96 h and 108 h, respectively, then remained stable or slightly decreased afterwards, whereas ACT production of *S. coelicolor* kept increasing and was approximately 2.5-fold higher than that of the *pptA* mutant at late time points (Figure 4b).

The in-frame *SLI_4383/pptA* deletion mutant was complemented by pMES18, an integrative and conjugative plasmid containing the coding sequence of *pptA* under the control of its own promoter. The integration of the intact *pptA* gene at the *attB* site of the chromosome in the apramycin-resistant ex-conjugants was confirmed by PCR. The complementation of this mutant strain led to a reduction of antibiotic production but did not restore a full non-producing phenotype similar to that of the wild type strain (Figure 4c). This might be due to the fact that the expression of *pptA* from its own promoter is weaker than when it is expressed from the promoter located upstream of *ppk*.

### 2.5. The Total Lipid Content of the SLI_4383/pptA Deletion Mutant Was Lower Than That of the Wild Type and ppk Mutant Strains of S. lividans

Since we previously demonstrated the existence of a negative correlation between total lipid content and antibiotic production in the *ppk* mutant of *S. lividans* and in *S. coelicolor* [5], we assessed the evolution of the lipid content of the *pptA* mutant throughout growth. Results shown in Figure 5 confirmed that the total lipid content of the *ppk* mutant was lower than that of the *wt* strain throughout the cultivation period but steadily increased with time. In contrast, the total lipid content of the *pptA* mutant that was at a similar level than that of the *ppk* mutant up to 72 h and decreased afterwards. As previously reported, the total lipid content of *S. coelicolor* was low and remained low throughout the cultivation period [5]. The lipid content as well as ACT production levels of the *pptA* mutant strain, being intermediary those of the *ppk* mutant and *S. coelicolor*, we concluded that the phosphate/energetic deficit of the *pptA* mutant was intermediary between that of the *ppk* mutant and that of *S. coelicolor*. These data confirmed the existence of a negative correlation between total lipid content and antibiotic production [5,24].

Interestingly, most of ACT produced by the three strains remained intracellular, up to 72 h but after this point most ACT produced by the *ppk* mutant and *S. coelicolor* was extracellular, whereas a large proportion of ACT remained intracellular in the *pptA* mutant even if this proportion decreased with time. To explain these differences, one can propose that ACT remains intracellular as long as it is needed within the cell to fulfill its putative anti-respiratory and anti-oxidant function [5,25]. In that respect, one cannot exclude than the surprisingly high level of ACT detected in the extracellular medium of *S. coelicolor* results from altered membrane permeability linked to its altered membrane composition [26] and/or from extensive cell lysis linked to severe phosphate limitation.

### 2.6. Proteins of the Pho Regulon Involved in Pi Supply and Saving Are Up-Regulated in the SLI_4383/pptA Mutant

Since antibiotic production usually occurs in condition of low phosphate availability, we carried out a comparative proteomic analysis of the *pptA* mutant with the *wt* strain of *S. lividans* to determine whether the *pptA* mutant suffers from phosphate stress. The results of this study, shown in Figure 6 a,b, revealed that the expression of the two-component system (TCS) PhoR/PhoP (SLI-4465/SLI-4466) as well as that of the regulatory protein PhoU (SLI_4464) was strongly up-regulated in the *SLI_4383* mutant as it was in the *ppk* mutant [16,17]. This TCS is known to control positively the expression of genes encoding proteins involved in phosphate scavenging, up-take, and saving in condition of Pi limitation [27]. Consistently, proteins belonging to the low affinity ABC phosphate transport system (PstSCAB), SLI_4381 (PstS), SLI_4380 (PstC), SLI_4379 (PtsA), and SLI_4378 (PstB), as well as proteins annotated as phosphatase (SLI_4434) or pyrophosphatase (SLI_3752) were clearly upregulated in the *pptA* mutant (Figure 6a).

Furthermore, proteins encoded by the neighboring genes of *pptA*, *ppk*, and *SLI_4382* were also upregulated in the mutant. At last, proteins known to be involved in the biosynthesis of phosphate free teichulosonic acid cell-wall glycopolymer (SLI_5146-SLI_5156) in replacement of phosphate containing teichoic acid, in condition of phosphate limitation [28], were also up-regulated in the *pptA* mutant (Figure 6b). Some of the proteins mentioned above were also up-regulated in the *ppk* mutant strain [29]. Altogether, these data indicated that the *pptA* mutant, as the *ppk* mutant suffers from phosphate limitation that resulted into energetic stress.

### 2.7. The ppk-pptA-SLI_4382 Region Is Highly Conserved in Streptomyces Species But Not in Other Prokaryotes

Out of 163 fully sequenced *Streptomyces* genomes from the RefSeq representative genomes database [30] that we analyzed using tblastn [31], we identified only 20 (12.3%) that did not have the *ppk-pptA-SLI_4382* gene arrangement found in *S. coelicolor*. In 7 of these genomes (4.3%), the *pptA* CHAD-encoding gene was missing and the *ppk* gene was immediately followed by the *SLI_4382* NUDIX hydrolase encoding gene. In the other 13 (8.0%), *SLI_4382* was missing but *pptA* was present downstream of *ppk*; in 2 of those cases, a likely *SLI_4382* orthologue was found elsewhere in the genome, far from *ppk*. No *Streptomyces* species were identified that lacked both *pptA* and *SLI_4382*, or *ppk*.

The genetic context of *ppk* in other prokaryotes was investigated using COGNAT [32], PSAT [33] and MicrobesOnline [34]. These analyses revealed that in the vast majority of genomes, *ppk* is not followed by a CHAD-encoding gene. Instead, the genes most commonly found downstream of *ppk* correspond to genes encoding PPX/GppA phosphatases [35], NUDIX hydrolases [36,37], and related enzymes such as MutT [38] and histidine phosphatases [39].

## 3. Discussion

Antibiotic biosynthesis has long been known to be triggered in condition of phosphate limitation that results in energetic stress and repressed in condition of phosphate proficiency. In consequence, polyphosphates (polyP) as ancient and ubiquitous phosphate polymers, specifically mobilized in condition of phosphate and energetic stress, might be involved in the regulation of antibiotic biosynthesis. We thus investigated the role of proteins encoded by genes of the *ppk-pptA-SLI_4382* operon, annotated as possibly involved in polyP metabolism, in the regulation of antibiotic biosynthesis in *S. lividans*.

This study and previous studies revealed that the expression of the two-component system PhoR/PhoP and of genes under its positive control, involved in Pi supply and saving, was strongly up-regulated in the *pptA* (Figure 6a,b) and *ppk* mutants [17]. This indicated that these mutants suffered from phosphate stress/low phosphate availability. The phosin PptA possesses a CHAD domain forming a positively charged tunnel thought to accommodate negatively charged polyP [19,20]. This polyP binding protein is thus most likely involved in the degradation of polyP to supply phosphate intracellularly. The simplest hypothesis would be that PptA acts as an exo-polyphosphatase (PPX-enzyme). However, no similarities nor conserved motifs could be found between PptA and any known exo-polyphosphatases and a purified protein related to SLI_4383/PptA originating from *S. chartreusis* [20] did not show any exo-polyphosphatase activity in vitro (data not shown). We thus propose that the function of PptA would be to maintain polyphosphates in a suitable conformation for their degradation into phosphate by PPX-like enzymes or by more general phosphatases. PptA would thus constitute an accessory factor for these enzymes. The generation of phosphate from polyphosphate would be totally impaired in the *pptA* mutant but would remain at a low level in the *ppk* mutant since *pptA* is weakly expressed from its own promoter in this strain. The phosphate limitation would thus be more severe in the *pptA* mutant than in the *ppk* mutant but far less severe in these two mutant strains than in *S. coelicolor*. Consistently, the *pptA* mutant produces ACT earlier and more abundantly than the *ppk* mutant but less abundantly than *S. coelicolor*.

The *pptA* mutant had a lipid content lower than that of the *ppk* mutant after 72 h but much higher than that of *S. coelicolor*. Interestingly, after 72 h, the total lipid content of the *ppk* mutant increased whereas that of the *pptA* mutant decreased. The reduced total lipid content of these mutant strains was most likely due to their lower phospholipid content due to low phosphate availability [26] as well as to the consumption of acetylCoA to support the activation of the oxidative metabolism of the strains to re-establish their energetic balance, as proposed previously for *S. coelicolor* [5,40]. The increase of the total lipid/phospholipid content of the *ppk* mutant after 72 h likely indicates enhanced phosphate availability, likely due to the expression of *pptA* from its own promoter taking place in this strain but not in the *pptA* deletion mutant.

Concerning the enhanced antibiotic production of these strains, we have previously reported that the great antibiotic production of *S. coelicolor* is, at least in part, linked to the weak expression in this strain, of the two-component system PhoR/PhoP that positively controls phosphate supply and saving [22]. In consequence, *S. coelicolor* is severely limited in phosphate and suffers from energetic stress. We proposed that the sensing of such stress triggers homeostatic processes that contribute to the re-establishment of the energetic balance of the strain [5,25]. One of these processes is a strong reduction of phospholipids biosynthesis in order to save phosphate [26]. Another process is the activation of the oxidative metabolism of the strain (its TCA cycle) that consumes acetylCoA and thus limits lipids biosynthesis and generates reduced co-factors whose re-oxidation by the respiratory chain governs ATP synthesis [5,40]. However, when Pi becomes too scarce, ATP cannot be generated and that is in this context that the production of the benzochromanequinone, Actinorhodin (ACT) is triggered. ACT was proposed to act as an electron acceptor to reduce respiration efficiency and thus ATP generation as well as oxidative stress when phosphate becomes severely limited [5,25]. The earlier and higher ACT production of the *pptA* mutant compared to the *ppk* mutant simply indicates that the internal phosphate availability of the *pptA* mutant is lower than that of the *ppk* mutant. This study thus unexpectedly revealed that the antibiotic over-producing phenotype and low lipid content of the *ppk* mutant might be due, at least in part, to the weak expression of *pptA* in this strain, resulting in a low intracellular phosphate availability.

Most importantly, this study emphasizes the importance of polyphosphate reserves and their degradation for the internal supply of phosphate to repress antibiotic biosynthesis in *Streptomyces*. The *ppk-pptA-SLI_4382* operon is widely conserved in *Streptomyces* species and deletion of this operon, or more specifically, of genes encoding PptA-like proteins, constitutes a promising strategy to trigger/enhance antibiotic production in *Streptomyces* species via the reduction of intracellular phosphate availability. The implementation of such simple strategy in various *Streptomyces* species may allow the discovery of most needed novel antibiotics previously too weakly produced to be characterized.

## 4. Materials and Methods

### 4.1. Strains, Plasmids, and Culture Conditions

The strains used in this study are listed in Appendix A. *S. coelicolor* M145 and *S. lividans* TK24 were obtained from the John Innes Institute, Norwich, UK. The *E. coli* strain DY330 was used for the disruption of genes carried on cosmids [41]. *E. coli* strains were grown in LB medium for plasmid isolation with 100 μg/mL ampicillin (Amp), 50 μg/mL apramycin (Apr) as final concentrations. YENB or SOB media were used to prepare electro-competent *E. coli* cells [42]. HT, TSB and YEME media used to routinely grow *Streptomyces* strains and SFM medium was used to prepare spores [43]. Apramycin (Apr) and thiostrepton (Tsr) were added at a final concentration of 30 μg/mL in HT medium and 50 μg/mL in R2YE medium. Neomycin (Neo) was added in HT medium at a final concentration of 20 μg/mL.

### 4.2. DNA Manipulations and Transformation of Streptomyces and E. coli Strains

Total genomic DNA extraction and transformation of *E. coli* and *Streptomyces* strains were performed according to standard procedures [24]. All the plasmids used in this study are listed in Appendix A. DNA fragments were purified from PCR or agarose gels using the NucleoSpin Gel and PCR Clean-Up kit (Macherey Nagel, Düren, Germany). Plasmid extraction was carried out using the NucleoSpin plasmid kit (Macherey Nagel, Düren, Germany). A cosmid library of *S. coelicolor* was obtained from the John Innes Institute (Norwich, UK).

### 4.3. Construction of KO Mutant Strains and Complementation of the SLI_4383/pptA Mutant Strain

PCR-based REDIRECT technology was used to disrupt *pptA* and *SLI_4382* [44]. The genes were replaced in the cosmid StD84 [45,46] by the *att3-aac* cassette (originating from pOSV234), yielding pMES*/pptA::aac* and pMES/*SLI_4382::aac*. These cosmids were introduced into *S. lividans* TK24 by transformation of protoplasts. Double recombinants (Apr^R^, Neo^S^) were detected. The replacement of the genes by the *aac* cassette was confirmed by Southern blotting and hybridization in comparison with the original strain (data not shown). In order to obtain the in frame deletion mutant of *pptA*, the *att3-aac* cassette of *pptA::att3-aac* mutant was excised in vivo as described previously [47], yielding the *∆pptA*
^IFD^ mutant.

In order to complement the *∆pptA*
^IFD^ mutant strain, a 1.5-kpb fragment containing the entire *pptA* gene with its putative promoter was amplified by PCR using genomic DNA from *S. lividans* as template and the primers listed in (Appendix A) and cloned into pGEM-T Easy resulting in pMES17. The 1.5-kpb *Eco*RI fragment from pMES17 was ligated into pSET152 digested with *Eco*RI generating pMES18. Plasmids pSET152 and pMES18 were then introduced into the *pptA*^IFD^ mutant by conjugal transfer from *E. coli* S17.1 (Appendix A). The integration of each plasmid into the genome was checked by PCR.

### 4.4. Southern Blot Analysis

First, 4 μg of digested DNA were loaded on each lane, electrophoresed on a 0.7% agarose gel, and transferred to Hybond-N+ (GE Healthcare, Chicago, IL, USA) according to the manufacturer’s recommendations. Membranes were pre-hybridized for 1 h at 65 °C and hybridized for 17 h. The probe used was a 3880-bp fragment containing *SLI_4383* and *SLI_4382* obtained by PCR on StD84 (Appendix A). This probe was labeled with [α-^32^P] dCTP by using the Megaprime DNA-Labeling system (GE Healthcare, Chicago, IL, USA). Membranes were then washed thoroughly in 0.2× SSC (1× SSC consists of 0.015 M sodium citrate and 0.15 M NaCl [pH 7.7]) containing 0.1% SDS at 65 °C. Autoradiography was performed by using Fischer scientific films at −80 °C, with intensifying screens STORM, for 1 to 5 h.

### 4.5. RNA Preparation

RNA was isolated from mycelia of *S. coelicolor* M145, *S. lividans* TK24, and of a *phoP* mutant derived from this strain [17], grown for 40 h at 28 °C on the solid R2YE medium with no KH_2_PO_4_ added but containing 1 mM free K_2_HPO_4_ from elements of the media (condition of phosphate limitation) or with 4 mM KH_2_PO_4_ added (5 mM final, condition of phosphate proficiency). In order to preserve RNA integrity, the mycelium was immediately frozen in liquid nitrogen in a solution containing denaturing guanidinium thiocyanate buffer RA1 (Macherey-Nagel, Düren, Germany), phenol-chloroform and ß-mercaptoethanol (a reducing agent). The cells were then lysed and homogenized in the presence of glass beads (diameter <106 µm) using a Fast-Prep apparatus (Savant Instruments, Telangana, India). Total RNA was purified using the Nucleospin RNA kit (Macherey-Nagel, Düren, Germany), according to the manufacturer’s instructions. To remove residual DNA, a DNAse TURBO™ treatment (Invitrogen, Carlsbad, CA, USA) was performed at 37 °C for 1 h and total RNA was purified with the Nucleospin RNA Clean-Up kit (Macherey-Nagel, Düren, Germany). The RNA concentrations were quantified using the Nanodrop 2000 spectrophotometer (Thermo Scientific, Waltham, MA, USA). The integrity of the RNAs was verified using the Agilent 2100 bio-analyzer with the eukaryote total RNA 6000 Nano assay (Agilent Technologies, Santa Clara, CA, USA).

### 4.6. RT-qPCR Experiments

Oligonucleotides used in this study were obtained from Operon and are listed in Appendix A. Control PCR amplifications were performed using Taq DNA Polymerase from QIAGEN (Hilden, Germany) or the GC-rich PCR system from Roche (Basel, Switzerland).

Primers used for RT-qPCR experiments were designed using the Primer-Blast tool from NCBI and the Primer Express 3.0 software (Life Technologies, Carlsbad, CA, USA). Specificity and the absence of multi-locus matching at the primer site were verified by BLAST analysis. Primers used for RT-PCR analysis are listed in Appendix A and schematic representation of their location is provided in Figure 2. The efficiency of each pair of primers was first checked by PCR on genomic DNA. The amplification efficiencies of primers pairs were determined using the slopes of standard curves obtained by a ten-fold dilution series. Amplification specificity for each real-time PCR reaction was confirmed by analysis of the dissociation curves. Each sample measurement was made in duplicate and four independent RNA biological samples were prepared for each condition. Determined Ct values were then exploited for further analysis. qPCR analysis in the absence of a reverse transcription step was performed on all RNA samples to check the absence of any DNA contamination.

The GenEx software (MultiD, Québec, QC, Canada) was used to select seven reference genes given the more constant level of expression in the various strains under study as well as in Pi limitation and proficiency. The geometric mean of the five most constant genes (*glk*/*SLI_2451*, *aspS*/*SLI_4040*, *gyrA*/*SLI_4129*, *gyrB*/*SLI_4131*, *rpoB*/*SLI_4926*) was used to normalize the data. *hrdB*/*SLI_6088* and *recG*/*SLI_5845* were excluded but *hrdB* was used for classical PCR and RT-qPCR control experiment shown in Figure 2c).

1 µg of total RNA was used as the template and reverse transcribed with the High Capacity cDNA Reverse Transcription kit (Life Technologies, Carlsbad, CA, USA) in the presence of RNase inhibitor and random primers following the manufacturer’s instructions, in a final volume of 20 µL. Quantitative PCR was performed on a Quant Studio 12K Flex Real-Time PCR System (Life Technologies, Carlsbad, CA, USA) with a SYBR green detection protocol. Next, 3 ng of cDNA were mixed with Fast SYBR Green Master Mix and a concentration of 750 nM of each primer in a final volume of 10 µL. The reaction mixture was loaded on 384 well microplates and submitted to 40 cycles of PCR (95 °C/20 s; [95 °C/1 s; 60 °C/20 s] × 40) followed by a fusion cycle to analyze the melting curve of the PCR products. The ΔΔCt method was used to determine the relative gene expression ratio with three biological replicates. The values of ΔΔCt of *S. lividans*, its *phoP* mutant and *S. coelicolor* were normalized and standardized by log transforming, mean centering and auto-scaling [48,49,50]. All data were subjected to the Student test and the results were presented as the mean of delta-delta-Ct ± standard deviation *p* < 0.05 was considered as statistically significant.

### 4.7. 5′. Rapid-Amplification-of-cDNA-Ends (5′ RACE) PCR

The transcriptional start sites of *SLI_4383/pptA* and *SLI_4382* were established by 5′ RACE PCR as described previously [51]. Briefly, 5 µg of total RNA isolated from a 48 h-old culture of *S. lividans* TK24 was individually used for reverse phase transcriptions with 2 picomoles of specific primers of *pptA*-GSP1 (Appendix A), using Superscript III first-strand synthesis supermix (Invitrogen, Carlsbad, CA, USA). After purification with a NucleoSpin Extract IIs column (Macherey-Nagel, Düren, Germany), the synthesized first-strand cDNA was added with a homopolymeric dA tail at the 3′ end using terminal deoxynucleotidyl transferase (Thermo Scientific, Waltham, MA, USA) according to the manufacturer’s instructions. The poly(dA)-tailed cDNA was again purified, and was then used as a template for the subsequent PCR amplification using a poly(dT) primer (70 homopolymeric T) and a second inner-gene-specific primer, *pptA*-GSP2 (Appendix A). An additional round of PCR was carried out using the original PCR product with a 1000-fold dilution as a template, with the poly(dT) primer and a nested *pptA*-GSP3 primer (Appendix A), to obtain a single specific band. The final PCR products were then sent for sequencing the nested *pptA*-GSP3 primer (Appendix A). The nucleotide immediately preceding the stretch of T residues was recognized as the transcriptional start site.

### 4.8. Assay of Extracellular and Intracellular Actinorhodin (ACT) Production

The strains *S. lividans ppk*::Ω*hyg*, *S. lividans pptA*^IFD^ deletion mutant, and *S. coelicolor* M145 were grown for 72 h at 28 °C. Each strain was grown on four individual 9 mm diameter plates (Figure 4b) or on four 5 mm diameter plates (Figure 4c) of solid R2YE medium with no KH_2_PO_4_ added but containing 1 mM free K_2_HPO_4_ from elements of the media (condition of phosphate limitation). To quantify extracellular ACT, at different points throughout growth, the cellophane of the 9 mm diameter plates was lifted and a agar cylinder of the growth medium was taken aseptically from each of the four replicates using an appropriate device (Figure 4b) or the totality of the agar medium present just under the cellophane disk cellophane of each of the 4 replicates was cut into small pieces (Figure 4c). The 4 agar cylinders together (Figure 4b) or the agar pieces (Figure 4c) were allowed to diffuse in 2 mL of water for 2 h at 4 °C. The first eluate was transferred into a new tube. The operation was repeated twice and 3 mL of HCl (3 M) were added to the 6 mL of final eluate. The mixture was incubated on ice for one night to allow ACT precipitation. To quantify intracellular ACT at different time points throughout growth (Figure 4b) or after 72 h of cultivation (Figure 4c), mycelium present in an area of 1 cm^2^ of each replicate (Figure 4b) or the totally of the mycelium present under the cellophane disk (Figure 4c), was scraped off, lyophilized, and weighted. Collected mycelium samples were incubated in 1 mL of KOH (1 M) for 30 mn at 4 °C under agitation. After centrifugation, the supernatant was transferred into a new tube and this operation was repeated five times. Two mL of HCl (3 M) were added to the 4 mL of final eluate. The mixture was incubated on ice one night to allow ACT precipitation. After centrifugation at 13,000× *g* for 30 mn at 4 °C and elimination of the supernatant, the reddish pellet of precipitated ACT was re-suspended in 1 mL KOH (1 M). The optical density of the KOH solutions corresponding to extra- and intra-cellular ACT was determined at 640 nm in a Shimadzu UV-1800 spectrophotometer using KOH (1 M) as blank. The Beer–Lambert law (molar attenuation coefficient of ACT = 25,320) was used to calculate the amount of ACT produced expressed in nano moles per mg of dry biomass [43]. For Figure 4c, the results obtained are presented as the mean ± standard error; a *p*-value < 0.05 was considered as statistically significant. The letters above the histograms indicate the significance of the differences. When two histograms bear two different letters that means that they are statistically significantly different (*p* > 0.05; Tukey adjusted comparisons). For Figure 4b, the standard error could not be presented since the 4 agar plugs, as the four mycelial samples, were incubated in the same tube.

### 4.9. Determination of Total Lipid Content Using Attenuated Total Reflectance-Fourier Transform Infra Red Spectroscopy (ATR-FTIRS) Measurements

In order to determine total lipid content of the strains grown in the same conditions as above (Figure 5), lyophilized mycelial samples were subjected to FTIR spectroscopy using a Bruker Vertex 70 FTIR spectrophotometer with a diamond ATR attachment (PIKE MIRacle crystal plate diamond ZnSe) and a MCT detector with a liquid nitrogen cooling system as described previously [52]. The total lipid content of the strains expressed first in arbitrary units, was converted into μg of fatty methyl esters (FAME) per mg of dry mycelium using the converting equation previously established [53,54].

### 4.10. Comparative Proteomic Analysis of the Wild Type Strain of S. lividans and of the ∆SLI_4383 Mutant

Total protein extracts of thirty two samples resulting from four biological replicates of *S. lividans* WT and its *SLI_4383* deletion mutant grown on R2YE limited (1 mM) and proficient (5 mM) in phosphate for 48 and 60 h, were alkylated before digestion by Lysyl-Endopeptidase (Wako Chemicals USA) and sequencing-grade-modified trypsin (Promega, Madison, WI, USA). The resulting proteolytic peptides were pre-cleaned, concentrated under vacuum, and stored before mass spectrometry analysis as described previously [22]. Trypsin-generated peptides were analyzed by nanoLC-MSMS using a nanoElute liquid chromatography system (Bruker, Billerica, MA, USA) coupled to a timsTOF pro mass spectrometer (Bruker, Billerica, MA, USA). Then, 1 µg of protein digest in 2 µL of loading buffer (2% acetonitrile and 0.05% trifluoroacetic acid in water) were loaded on an Aurora analytical column (ION OPTIK, 25 cm × 75 µm, C18, 1.6 µm) and eluted with a gradient of 0–35% of solvent B for 100 min. Solvent A was 0.1% formic acid and 2% acetonitrile in water, and solvent B was 99.9% acetonitrile with 0.1% formic acid. MS and MS/MS spectra were recorded from *m/z* 100 to 1700 with a mobility scan range from 0.6 to 1.5 V·s/cm^2^. MS/MS spectra were acquired with the PASEF (parallel accumulation—serial fragmentation (PASEF)) ion mobility-based acquisition mode using a number of PASEF MS/MS scans set as 10. MS and MSMS raw data were processed and converted into mgf files with DataAnalysis software (Bruker, Billerica, MA, USA).

Proteins Identifications were performed using the MASCOT search engine (Matrix science, London, UK) against the *S. coelicolor* and *S. lividans* protein databases from UniprotKB (15012020). Database searches were performed using trypsin cleavage specificity with two possible miscleavages. Carbamidomethylation of cysteines was set as fixed modification and oxidation of methionines as variable modification. Peptide and fragment tolerances were set at 25 ppm and 0.05 Da, respectively. Proteins were validated when identified with at least two unique peptides. Only ions with a score higher than the identity threshold and a false-positive discovery rate of less than 1% (Mascot decoy option) were considered.

Mass spectrometry based-quantification was performed by label-free quantification using either spectral count (SC) or MS1 ion intensities named XIC (for eXtracted Ion Current). For the first method, total MS/MS spectral count values were extracted from Scaffold software (version Scaffold_4.11.1, Proteome software Inc., Portland, OR, USA), filtered with 95% probability and 1% FDR for protein and peptide thresholds, respectively. For the MS1 ion intensity-based method, MS raw files were analyzed with Maxquant software (v 1.6.10.43) using the maxLFQ algorithm with default settings and enabled 4D feature alignment for a more specific quantitative analysis (match between run function including CCS alignment). Normalization was set as default and identifications with Andromeda were performed using the same search parameters as MASCOT.

Statistical quantitative analyses were based on two different generalized linear models depending on the type of data. The discrete SC (1) and continuous XIC (2) abundances values were modeled respectively as following:

(1) SC = μ + strain + medium + time + replicate + strain x medium + strain x time + medium x time + strain x medium x time + ε ~ Pois(λ);

(2) Log2(XIC) = μ + strain + medium + time + replicate + strain x medium + strain x time + medium x time + strain x medium x time + ε ~ N(0, σ).

Terms represent effect of different conditions and their interactions on protein abundance. Effects were estimated by maximum likelihood and statistical significances were calculated using likelihood ratio test based on the analysis of deviance. *p*-values were adjusted using the Benjamini–Hochberg procedure for multiple testing correction. For each protein, a significant difference in abundance for the 66 combinations of pairwise comparisons was set at 10 for SC and 1 log fold change for XIC. A threshold of 10 significant pairs and an adjusted *p*-value of 0.05 was used to consider a protein as significantly variable. All the statistical analyses were performed by a homemade R script.

### 4.11. Computer Analysis of Protein Sequences

Sequence analysis and comparison were performed with various programs including BLAST [55], GeneMark.hmm [56,57], EasyGene [58,59], and AMIGene [60]. The conserved domain database [61] was used to detect known functional domain.

## Figures and Tables

**Figure 1 antibiotics-10-00325-f001:**
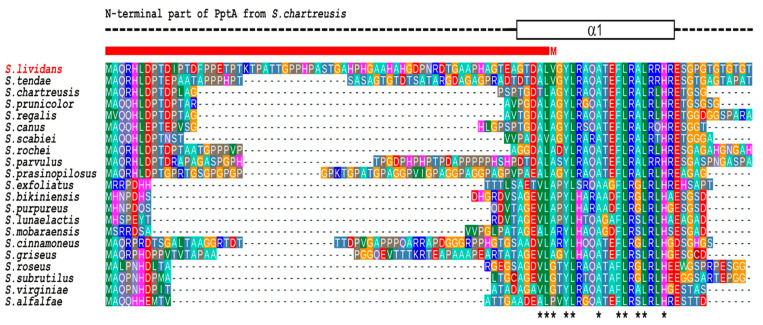
Multiple sequence alignment of the N-terminal region of PptA-like proteins from various *Streptomyces*. The crystal structure of *S. chartreusis* PptA (PDB entry 6RN5) is schematically shown above the alignment. Dashed line represent structurally disordered areas, whereas the rectangle in solid lines represent the N-terminal α-helix (α1). Asterisks under the alignment indicate strongly conserved hydrophobic residues expected to play key roles in the stability of the protein fold. The red bar above the alignment indicates the region that is missing in the current database entry for PptA (SLI_4383) of *S. lividans* (GenBank entry EFD67904.1) due to incorrect assignment of the translation start site (M).

**Figure 2 antibiotics-10-00325-f002:**
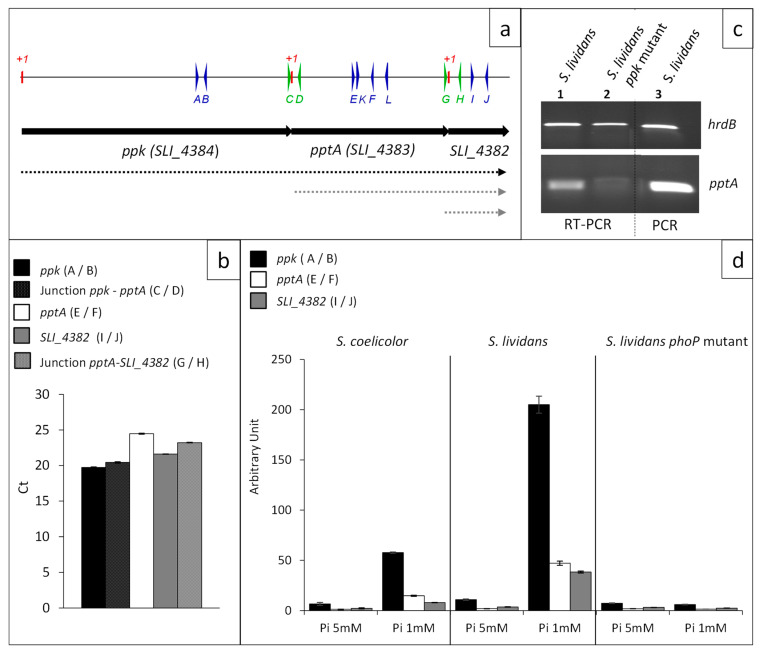
Transcriptional situation of the contiguous *ppk*, *pptA*, and *SLI_4382* genes. (**a**) Schematic representation of the genetic map of the region encompassing genes *ppk*, *pptA*, and SLI_4382 represented by black arrows. Primers used to establish the transcriptional situation are represented by thin vertical arrows on a line above the genetic map. Deduced transcriptional situation is represented by dotted lines below the genetic map (**b**) Level of expression of *ppk*, *pptA*, and *SLI_4382* and their junctions determined by RT-qPCR with RNA prepared from *S. lividans* and its *phoP* mutant grown in conditions of Pi limitation. Position of the primers used is shown in (**a**) and sequences of the primers are listed in Appendix A. The primers are either internal to *ppk* (A/B, plain black histograms), *pptA* (E/F, plain white histograms), and *SLI_4382* (I/J, plain grey histograms) or encompass the *ppk*/*pptA* junction (C/D, dotted black histograms) or the *pptA*/*SLI_4382* junction (I/J, dotted grey histograms). (**c**) Determination by RT-PCR of the expression of *pptA* in *S. lividans* WT (lane 1) and in the *ppk* mutant of this strain (lane 2) using primers K and L. Control of DNA amplification using the same primers is shown in lane 3 (**d**) Level of expression of *ppk* (black histograms), *pptA* (white histograms), and *SLI_4382* (grey histograms) determined by RT-qPCR with RNA prepared from *S. coelicolor*, *S. lividans*, and the *phoP* mutant of *S. lividans* grown on R2YE in condition of Pi limitation (1 mM) and proficiency (5 mM).

**Figure 3 antibiotics-10-00325-f003:**
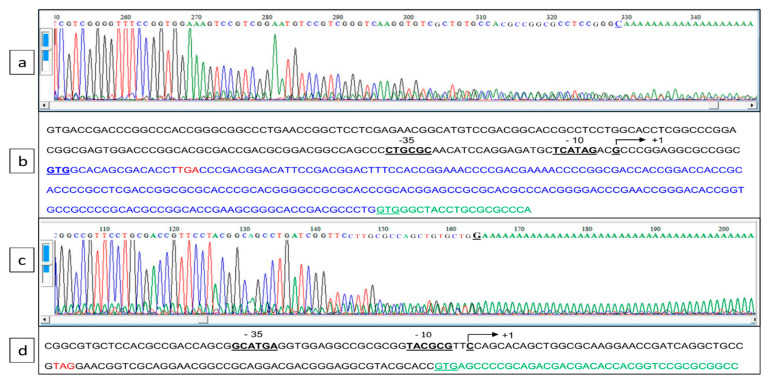
Determination of the transcriptional start point of *SLI_4383/pptA* and *SLI_4382* established by 5′ RACE PCR. (**a**,**c**) Chromatograms of 5′ RACE PCR of the *pptA* and *SLI_4382* promoter region, respectively. (**b**,**d**) Annotated sequence of the promoter region of *pptA* and *SLI_4382*, respectively. The bent arrow represents the transcriptional start site, putative −10 and −35 boxes are underlined, the translational start codons (GTG) are in bold letters and underlined, and the translational stop codon of *ppk* and *SLI_4383* are in red. The erroneously annotated GTG start codon of PptA in the GenBank entry for the *S. lividans* genome (GG657756.1) is underlined.

**Figure 4 antibiotics-10-00325-f004:**
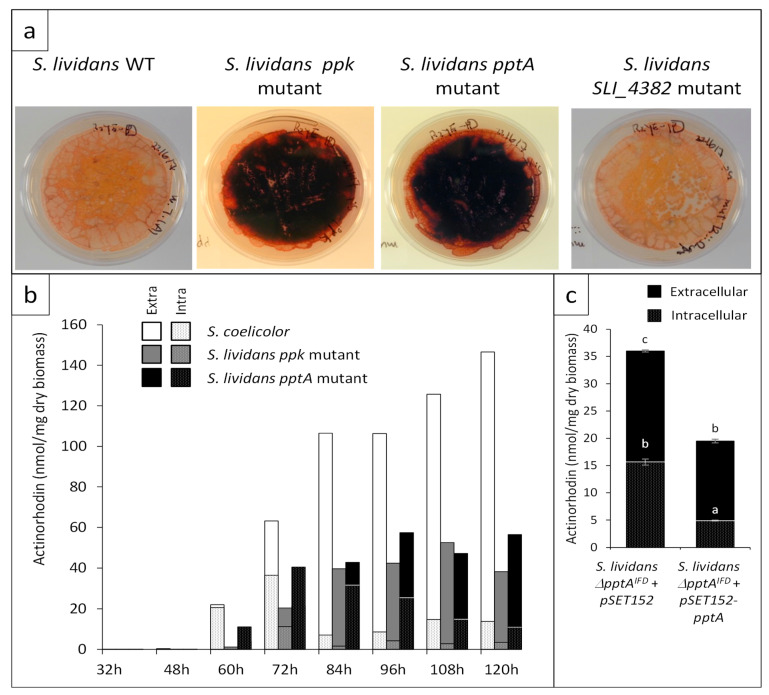
Antibiotic production of *S. lividans* TK24 wild type and of the *ppk*, *pptA*, and *SLI_4382* mutants as well as of the complemented *SLI_4383* mutant (**a**) Pictures of lawns of the strains grown on the surface of a cellophane disk laid onto plates of R2YE solid medium limited in phosphate (1 mM) after 96 h of incubation at 28 °C. (**b**) Assay of extracellular (plain section) and intracellular (dotted section) actinorhodin production of *S. coelicolor* (white histograms) and the *ppk* (grey histograms) and *pptA* (back histograms) mutants of *S. lividans* at various time points throughout cultivation. (**c**) Complementation of the *pptA* mutant by pSET152-*pptA* (pMES18). Assay of extracellular (plain black histograms) and intracellular (dotted black histograms) actinorhodin production of the *pptA* mutant containing pSET152 (control) or pSET152-*pptA* (pMES18) after 72 h of cultivation.

**Figure 5 antibiotics-10-00325-f005:**
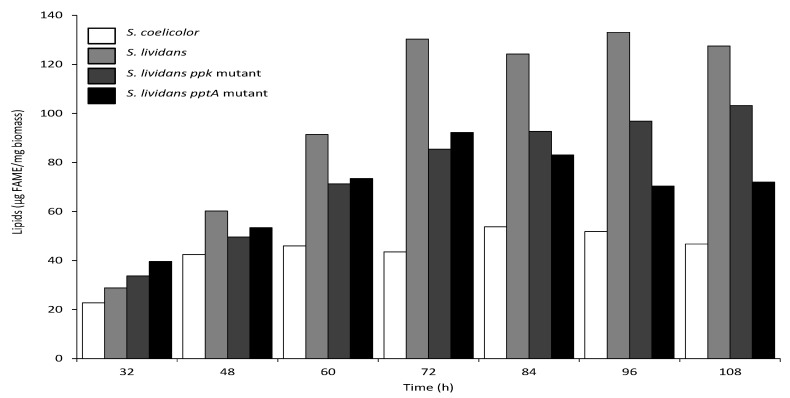
FTIRS analysis of total lipid/fatty methyl ester (FAME) content of *S. coelicolor* (white histograms), *S. lividans* (light grey histograms), *S. lividans ppk* mutant (dark grey histograms), and *S. lividans SLI_4383/pptA* mutant (black histograms) at different points throughout the cultivation period carried out on solid R2YE medium with glucose (50 mM) as the main carbon source and containing 1 mM phosphate (condition of phosphate limitation) at 28 °C.

**Figure 6 antibiotics-10-00325-f006:**
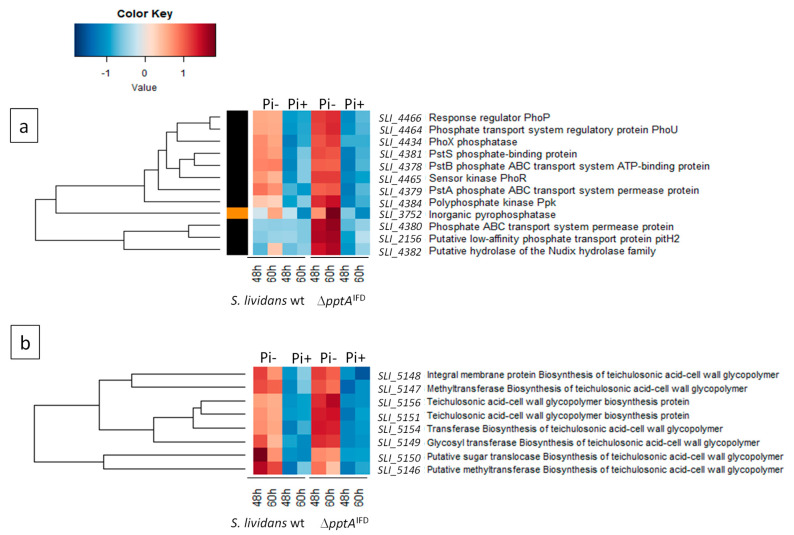
Heatmap representation of proteins of the Pho regulon involved in phosphate scavenging and up-take (**a**) as well as in the biosynthesis of phosphate free teichulosonic acid biosynthesis in replacement of teichoic acids (**b**) with significant abundance change (ANOVA, adjusted *p* value < 0.05) between *S. lividans* (TK24) and its *SLI_4383/pptA* deletion mutant. The strains were grown at 28 °C for 48 and 60 h on solid R2YE medium containing glucose (50 mM) as main carbon source and 5 mM (Pi+) or 1 mM (Pi-) phosphate (conditions of phosphate proficiency or limitation, respectively). Protein identifiers are indicated as SLI numbers for both strains and by predicted functions.

## Data Availability

RT-qPCR data are presented in the paper in Figure 2 and row data (Excell file) can be obtained from the corresponding author on request. Proteomic data are presented in the paper in Figure 6 and row data (Excell file) can be obtained from the corresponding author on request.

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
