# Peer review of "The Phosin PptA Plays a Negative Role in the Regulation of Antibiotic Production in Streptomyces lividans"

_antibiotics, 2021, doi:10.3390/antibiotics10030325_

Round 1
Reviewer 1 Report
The manuscript reports on the discovery and characterization of a novel gene (SCO4144) involved in the control of antibiotic production in Streptomyces lividans. SCO4144, encoding a putative polyphosphate binding protein was shown to be cotranscribed with the polyphosphate kinase gene ppk, well characterized by this group in previous work. phoRP and other genes of the pho regulon were upregulated in the SCO4144 mutant, suggesting phosphate stress which induces actinorhodin production.
The manuscript is sound and well written. I only have minor comments.
- Confusing nomenclature: SCO4144 refers to a gene from S. coelicolor not to its homologue from S. lividans. The correct designation is probably “SLI_4383”. In other publications from this group, the homologous protein from S. chartreusis was named “phosin” and PptA (Polyphosphate targeting protein). Renaming “SCO4144” into PptA would solve the confusing and inappropriate naming.
- Mutant construction: If I understood correctly, the authors used a S. coelicolor cosmid (StD84) to replace sco4144 and used the respective cosmid to replace the corresponding region in the S. lividans genome. Since the exact position of homologous recombination cannot be predicted and since S. lividans and S. coelicolor are not 100 % identical, several additional “mutations” could be present in the S. lividans sco4144 mutant. Genetic complementation of the mutant suggests that the conclusions on the mutant phenotype might be correct. Nevertheless, the authors should comment on the similarity (additional genes present/absent?) of cosmid StD84 to the respective S. lividans region.
- Line 116-119: …revealed the presence of a highly conserved N-terminal extension in these orthologues but not in SCO4144 (Figure 1). From Fig. 1, I do not understand what is meant.
- Line 215-217, Effect of SCO4144 on antibiotic production. Figure 4. Assay of antibiotic production. Like S. coelicolor, S. lividans encodes four distinct BGCs. Since the authors analysed only actinorhodin, a general effect of SCO4144 on antibiotic production is not supported.
- Line 513-532. Assay of extracellular and intracellular ACT production. Methodology only describes the determination of extracellular Act concentration. Can the intracellular accumulation of Act in the SCO4144 mutant (line 264/5) also be caused by a more acidic intracellular pH (making Act non-soluble), compared to that of the ppK mutant or S. coelicolor?
- References are not given in alphabetical order
Author Response
ANSWER TO REVIEWER COMMENTS
We wish to thank the reviewer for their constructive comments. Most modifications recommended were done and the changes are highlighted in yellow in the new version of the manuscript.
Reviewer 1
The manuscript reports on the discovery and characterization of a novel gene (SCO4144) involved in the control of antibiotic production in Streptomyces lividans. SCO4144, encoding a putative polyphosphate binding protein was shown to be cotranscribed with the polyphosphate kinase gene ppk, well characterized by this group in previous work. phoRP and other genes of the pho regulon were upregulated in the SCO4144 mutant, suggesting phosphate stress which induces actinorhodin production.
The manuscript is sound and well written. I only have minor comments.
- Confusing nomenclature: SCO4144 refers to a gene from S. coelicolor not to its homologue from S. lividans. The correct designation is probably “SLI_4383”. In other publications from this group, the homologous protein from S. chartreusis was named “phosin” and PptA (Polyphosphate targeting protein). Renaming “SCO4144” into PptA would solve the confusing and inappropriate naming.
This was changed as requested by the reviewer.
- Mutant construction: If I understood correctly, the authors used a S. coelicolor cosmid (StD84) to replace sco4144 and used the respective cosmid to replace the corresponding region in the S. lividans genome. Since the exact position of homologous recombination cannot be predicted and since S. lividans and S. coelicolor are not 100 % identical, several additional “mutations” could be present in the S. lividans sco4144 mutant. Genetic complementation of the mutant suggests that the conclusions on the mutant phenotype might be correct. Nevertheless, the authors should comment on the similarity (additional genes present/absent?) of cosmid StD84 to the respective S. lividans region.
We checked the genetic organization of a 30 kb region upstream and downstream of ppk in S. lividans and S. coelicolor and found that it was identical in the two species. Furthermore, very few “non-mutagenic” nucleotides changes were noticed.
Line 116-119: …revealed the presence of a highly conserved N-terminal extension in these orthologues but not in SCO4144 (Figure 1). From Fig. 1, I do not understand what is meant.
We modified the figure and its comments to improve clarity.
- Line 215-217, Effect of SCO4144 on antibiotic production. Figure 4. Assay of antibiotic production. Like S. coelicolor, S. lividans encodes four distinct BGCs. Since the authors analysed only actinorhodin, a general effect of SCO4144 on antibiotic production is not supported.
We changed antibiotics by actinorhodin even if it is most likely that, as demonstrated for the ppk mutant, the pptA mutant has also a positive effect on CDA and RED production.
- Line 513-532. Assay of extracellular and intracellular ACT production. Methodology only describes the determination of extracellular Act concentration.
- The protocole to assay intracellular ACT production is provided in the new version.
Can the intracellular accumulation of Act in the SCO4144 mutant (line 264/5) also be caused by a more acidic intracellular pH (making Act non-soluble), compared to that of the ppK mutant or S. coelicolor?- That is a question difficult to answer. We do not think that the intracellular pH of the pptA mutant is more acidic than that of the ppk mutant but we cannot exclude it either. We think that ACT stays longer inside the pptA mutant because its presence inside is needed since Pi limitation and its consequences are more severe in this strain than in the ppk mutant.
- References are not given in alphabetical order
The references are in the format recommended by the journal.
Reviewer 2 Report
The work results investigates a role of two novel genes/proteins of the ppk operon in the regulation of phosphate-starvation and related secondary metabolism increase. The experiments were properly designed and data presented and interpreted correctly. The work data can be directly applied in the design of heterologous producer strains, activation of silent biosynthetic gene clusters experiments in general and so on.
I have just minor points and questions to the authors:
- Sometimes, references are improperly cited within the article - multiple references not included in a single brackets. - e.g. lines 48 and 52
- Ppk abbreviation should be explained as polyphosphate kinase when first mentioned in the text - line 24. The same applies to ACT- actinorhodin - line 222.
- Why you haven't included S. coelicolor SCO4144 sequence in the alignment in Figure 1? - line 120
- I think that using "SCOxxxx" annotation also for genes/proteins of S. lividans is slightly misleading for the reader as these link to S. coelicolor. Either "SCOxxxx homologue" or corresponding "SLIVxxxx" should be used in protein /gene identification in S. lividans. Especially when authors work with both strains.
- Why was not the ppk mutant lacking the resistance cassette prepared to verify the polar effect of the mutation - line 173?
- "The overlap of" rather than "The overlapping between" - line 211
- Are the growth rates of mutants the same/comparable (or at least the biomass at the time point of ACT production assessment? - line 229
- In the Discussion authors suggest sco4144 to operate on the proper polyP conformation for the degradation. Is not it possible that the protein products serves as a receptor for polyP concentration sensing and the function is solely regulatory, e.g. via conformation-directed protein interactions? - line 347
- "Two-component system" rather than "two components system" - line 368
- A reference missing for YENB and SOB media. - line 403
- "750 nM of each primer" in the line 486 - does it refer to molar concentration or amounts of the primer? Consider rephrasing to make it clear.
- Replace "pico mole" with "picomole". - line 500
- Replace "KOH 1 M" with "1 M KOH" or "KOH (1 M)". - lines 529 and 530
- Remove "(2017)"- as there are three references linking also to a work from 2016. - line 542
- Italics missing in latin names of microbes in most of the references - e.g. lines 651, 652, 657, 662, 663, 666 and following. Please correct.
Author Response
ANSWER TO REVIEWER COMMENTS
We wish to thank the reviewers for their constructive comments. Most modifications recommended were done and the changes are highlighted in yellow in the new version of the manuscript.
Reviewer 2
The work results investigates a role of two novel genes/proteins of the ppk operon in the regulation of phosphate-starvation and related secondary metabolism increase. The experiments were properly designed and data presented and interpreted correctly. The work data can be directly applied in the design of heterologous producer strains, activation of silent biosynthetic gene clusters experiments in general and so on.
I have just minor points and questions to the authors:
- Sometimes, references are improperly cited within the article - multiple references not included in a single brackets. - e.g. lines 48 and 52
This was corrected
- Ppk abbreviation should be explained as polyphosphate kinase when first mentioned in the text - line 24. The same applies to ACT- actinorhodin - line 222.
This was corrected
- Why you haven't included coelicolor SCO4144 sequence in the alignment in Figure 1? - line 120
We have not included SCO4144 S. coelicolor sequence because it is identical to that of S. lividans.
- I think that using "SCOxxxx" annotation also for genes/proteins of lividans is slightly misleading for the reader as these link to S. coelicolor. Either "SCOxxxx homologue" or corresponding "SLIVxxxx" should be used in protein /gene identification in S. lividans. Especially when authors work with both strains.
This was corrected
- Why was not the ppk mutant lacking the resistance cassette prepared to verify the polar effect of the mutation - line 173?
When the ppk mutant was constructed, the excision tools did not exist but at that time we checked the expression of pptA in the ppk mutant using Northerm blot and could detect a transcript.
- "The overlap of" rather than "The overlapping between" - line 211
This was corrected
- Are the growth rates of mutants the same/comparable (or at least the biomass at the time point of ACT production assessment? - line 229
The growth rate of the ppk and pptA mutants are similar and slightly slower than that of the wt strain of S. lividans in condition of Pi limitation but ACT production being expressed in nanomole per mg of dry biomass, the values can be compared.
- In the Discussion authors suggest sco4144 to operate on the proper polyP conformation for the degradation. Is not it possible that the protein products serves as a receptor for polyP concentration sensing and the function is solely regulatory, e.g. via conformation-directed protein interactions? - line 347
That is difficult to answer this comment. The function proposed for PptA seems the more likely to us but other functions could indeed be envisaged.
- "Two-component system" rather than "two components system" - line 368
This was corrected
- A reference missing for YENB and SOB media. - line 403
The reference below was added.
23 Sharma, R C; Schimke, R T Preparation of electro competent E. coli using salt-free growth medium. BioTechniques 1996, 20(1) 42–44.
- "750 nM of each primer" in the line 486 - does it refer to molar concentration or amounts of the primer? Consider rephrasing to make it clear.
Yes it does refer to the molar concentration. To clarify the sentence was slightly modified as “a concentration of 750 nM of each primer in a final volume of 10 µL.”
Replace "pico mole" with "picomole". - line 500
Correction done
- Replace "KOH 1 M" with "1 M KOH" or "KOH (1 M)". - lines 529 and 530
Modification done
- Remove "(2017)"- as there are three references linking also to a work from 2016. - line 542
That is better to maintain the three references.
- Italics missing in latin names of microbes in most of the references - e.g. lines 651, 652, 657, 662, 663, 666 and following. Please correct.
The references are in the format recommended by the journal.